# Integrating eHealth within a Transforming Mental Healthcare Setting: A Qualitative Study into Values, Challenges, and Prerequisites

**DOI:** 10.3390/ijerph181910287

**Published:** 2021-09-29

**Authors:** Karin Lorenz-Artz, Joyce Bierbooms, Inge Bongers

**Affiliations:** 1Tranzo, Tilburg School of Social and Behavioral Sciences, Tilburg University, 5000 LE Tilburg, The Netherlands; j.j.p.a.bierbooms@tilburguniversity.edu (J.B.); i.m.b.bongers@tilburguniversity.edu (I.B.); 2Mental Health Care Institute Eindhoven, 5626 ND Eindhoven, The Netherlands

**Keywords:** eHealth, online treatment, open dialogue, transformation, mental health care, client-centered healthcare, network-oriented healthcare

## Abstract

Mental health care is shifting towards more person-centered and community-based health care. Although integrating eHealth within a transforming healthcare setting may help accomplishing the shift, research studying this is lacking. This study aims to improve our understanding of the value of eHealth within a transforming mental healthcare setting and to define the challenges and prerequisites for implementing eHealth in particular within this transforming context. In this article, we present the results of 29 interviews with clients, social network members, and professionals of an ambulatory team in transition within a Dutch mental health care institute. The main finding is that eHealth can support a transforming practice shifting towards more recovery-oriented, person-centered, and community-based service in which shared-decision making is self-evident. The main challenge revealed is how to deal with clients’ voices, when professionals see the value of eHealth but clients do not want to start using eHealth. The shift towards client-centered and network-oriented care models and towards blended care models are both high-impact changes in themselves. Acknowledging the complexity of combining these high-impact changes might be the first step towards creating blended client-centered and network-oriented care. Future research should examine whether and how these substantial shifts could be mutually supportive.

## 1. Introduction

A paradigmatic shift is underway in mental health care focusing on empowering clients and their environment and enabling personal recovery, rather than stabilization and symptom reduction as a clinical outcome [1,2,3,4,5]. Moreover, access to and continuity of care and service quality have to be approved to meet the growing number of people facing psychological difficulties [1,5,6]. Health care services need to become more recovery-oriented, person-centered, and community-based in which shared-decision making is a matter of course [3,6,7,8,9]. This central notion of empowerment is profound and complex and cannot be downsized to an expert giving power to a client [10]. Clients, being persons in treatment with a long-term mental illness, often are familiar with their mental vulnerabilities and gain their own experience-based expertise [11]. Through reflection and dialogue, empowerment can ultimately only be achieved by clients themselves [12]. This means, e.g., that the clinician should be a guide in this process rather than an expert (guiding the client in making treatment decisions, rather than knowing what is best for a client). At the same time, the client needs to change from a passive listener to an active participant in the treatment process (participating in treatment and decision-making rather than only listening to what the expert says) [10]. 

Digitalization, such as the use of eHealth, may help with this transformation [3,5,6,9,13]. eHealth’s ability to make this contribution to the transformation lies within the fact that the term eHealth, in a broader sense, “characterizes not only a technical development, but also a state-of-mind, a way of thinking, an attitude, and a commitment for networked, global thinking, to improve health care locally, regionally, and worldwide by using information and communication technology” [14] (p. 1). Namely, eHealth could help healthcare models shift from traditional client-clinician roles into more person-centered and community-based services where clients are empowered and contribute to making shared decisions [5,6,9,10,13,15]. Moreover, the use of a personal health record, as a digital platform, could facilitate direct communication between the client and his/her social network members [16]. Such a platform enables the continuous participation of the clients in their own care even when they cannot meet in person. In this manner, eHealth may also solve the logistical challenges of shifting towards community-based healthcare [1,17], because planning the treatment within a community-based practice demands more coordination than meeting with the client alone [16]. Furthermore, it may improve collaboration between service providers [17]. In addition, eHealth increases accessibility to and the scope of healthcare services [13,18,19,20,21]. 

Many eHealth implementation studies are fragmented across multiple subspecialty areas [22,23], focusing on, e.g., particular eHealth tools, e.g., [21] or specific client populations such as Schizophrenia Spectrum Disorders, e.g., [24]. These studies indicate that the implementation of eHealth itself seems to be complex [25,26]. Circling back to the paradigmatic shift in mental health care, the question is, what eHealth implementation entails within a substantial transforming context. To our best knowledge, no research specifically reports on the implementation of eHealth within a transforming mental health care setting shifting towards more person-centered and community-based service. It is believed that digitalization, such as the use of eHealth, may help with this transformation [3,5,6,9,13], but it remains to be researched whether this transforming context is helpful or more challenging for the implementation of eHealth. 

This study aimed to help the transition towards person-centered and community-based care models by improving our understanding of the value of eHealth within such a transforming mental healthcare setting and to define the challenges and prerequisites for implementing eHealth, in particular within this transforming context. 

## 2. Materials and Methods

### 2.1. Context

This study took place within a pilot project, setting up a multidisciplinary ambulatory Open dialogue (OD) team within GGz Eindhoven and the Kempen (GGzE), a Dutch mental health care institution based in the South of the Netherlands. This multidisciplinary pilot team has approximately 285 clients, with approximately 20–30 new clients being referred every year. OD clients, eligible for treatment within this pilot, are adults, suffer from severe mental illness associated with severe limitations in social and societal functioning for at least one year, and need treatment within a coordinated network of professionals with different expertise.

Open dialogue (OD)—worldwide implemented in several countries—is an example of the aforementioned transforming healthcare service [27]. OD is an innovative person-centered, network-oriented healthcare model within the biological-driven field of psychiatry. As well as giving a different perception of mental health problems [4], OD radically reorganizes the treatment system as a whole [28,29]. OD organizes care and therapeutic interventions so that primary treatment involves meetings with clients and their social and professional networks. The dialogical process, including the multiple viewpoints from the client’s entire network, is the innovative core of OD in psychiatry [30]. The implementation of OD into everyday practice to achieve a more person-centered healthcare model is challenging [31]. 

The OD pilot team was formed in 2017, after completing their OD training in England, and consists of professionals from different ambulatory teams. The pilot had two main aims. The first aim was to adhere to the seven OD principles (see Table 1). The seven OD principles form the backbone of OD practice and delineate the basis of the treatment organization and therapeutic stance [32]. The second aim was to explore how eHealth could be integrated into and contribute to OD practice. This research is related to the second aim: the exploration of the use of eHealth within this transforming practice.

The pilot OD team has access to several eHealth tools, including multi-function online treatment programs (including functionalities such as video sessions, messaging, modules with information and assignments, and social support network) and apps such as the multi-function messaging app, Ecomap app (mapping a social and professional network), and Mysolution app (direct access to personal solutions during stressful situations). These eHealth tools were introduced and applied to OD practice in the same manner as other treatment interventions. The value of eHealth within the OD practice was in the pilot explored with an eHealth expert who organized group sessions with OD professionals, clients, and their network members. Fifteen group sessions were organized every 4 to 6 weeks between December 2017 and March 2019. The aim of these group sessions was to show OD professionals, clients, and network members how to use eHealth tools and to discuss their needs, expectations, and experiences with eHealth. All OD professionals, clients, and social network members could participate (nobody was excluded). On average, seven members participated per session (four OD professionals and three clients, and sometimes a social network member). After each group session, the eHealth expert shared reports of the group’s session with the participants.

### 2.2. Design

This is a qualitative practice-oriented field study that took place within a pilot project, setting up a multidisciplinary ambulatory Open dialogue (OD) team and exploring the use of eHealth within this transforming setting. We interviewed clients, their social network, and OD professionals about their expectations and experiences with eHealth within the OD mental health care setting. This helped us to understand different perspectives, analyze individual perspectives and compare perspectives [36]. The study was approved by the Dutch Ethical Review Board of Tilburg School of Social and Behavioral Sciences, Tilburg University (REF EC-2018.91).

### 2.3. Participants and Recruitment

The team members of the OD team have their own caseload. The researcher asked each team member to invite all their clients to participate in this study by sharing information about the study. The researcher emphasized that all clients, including clients with a positive and negative or neutral attitude, are eligible. Eligible participants had a variety of mental health problems (diagnosed with, e.g., Schizophrenia, bipolar disorder, depression, or personality disorder), were at different stages of recovery, were not in a psychological crisis, and had different levels of motivation regarding receiving treatment and asking for help. All clients interested in participation received an information letter from the researcher and were asked to respond within two weeks. Clients were also asked to invite social network members. If clients were willing to invite their network for the interview, network members were asked to participate by the clients themselves. Once the clients signed the informed consent form, the interviews were planned. The clients could choose the location for the interview as long as the environment was quiet enough. Network members who signed up also participated in the interview with the client. Some interviewees also participated in the eHealth group sessions (see Context). 

With the intention to incorporate all perspectives, OD team members were recruited through purposive sampling with maximum variation in professional background, attitudes, expectations, and experience with eHealth [37]. The OD manager made a list of OD professionals that ensured maximum variation on these features. The OD manager and eHealth expert also participated and were referred to as OD professionals to ensure their anonymity. The OD professionals on the provided list, the OD manager, and the eHealth expert received a letter from the researcher, giving information about the study, inviting them to participate, and asking for a response within two weeks. All invited participants signed an informed consent form, after which the interviews were planned. Interviewees also participated in the eHealth group sessions. 

### 2.4. Data Collection 

In total, 29 open interviews of 1–1.5 h were conducted between December 2017 and March 2019. The topic list was used as a memory aid for the researcher during the open interviews, to ensure all relevant topics to answer the research question were covered. A topic list was constructed with several themes related to the research question and based on iterative steps of an implementation process, e.g., [38,39]: e.g., the value of eHealth, the manner in which eHealth is offered and used, involvement, received support, experienced struggles, and requirements. 

#### 2.4.1. Client/Network 

Ten interviews were conducted between December 2017 and March 2019 with ten clients and two network members. Clients could choose for an individual interview, with their social network or with support from a health care professional. All ten clients were adults, suffering from severe mental illness associated with severe limitations in social and societal functioning for at least one year, and needed treatment within a coordinated network of professionals with different expertise. Six interviews were held in the client’s home situation, and four were held on-site at the mental health care organization. Eight interviews were held individually with the client, and in two interviews a social network member was included next to the client (with a partner and a mother). 

#### 2.4.2. OD Professionals

Two rounds of semi-structured interviews were held with OD professionals: the first between December 2017 and June 2018 and the second between January 2019 and March 2019. The first round of interviews comprised eight individual interviews with the following OD professionals: one peer worker, six case managers, and one manager. One interview was held with two clinicians together at the interviewees’ request. The same OD professionals took part in the second round of interviews, except for the peer worker. Another peer worker was interviewed instead because he had more experience with eHealth. The first interviews with these OD professionals focused on their expectations of eHealth, and the second interviews on their experiences with eHealth. Both interviews covered the value of eHealth to OD practice as well as the challenges and prerequisites of implementing eHealth within OD practice.

#### 2.4.3. eHealth Expert

At the end of the project (March 2019), one semi-structured interview was held with the eHealth expert. This interview covered, in addition to the topics included in the group sessions, the value of eHealth to OD practice as well as the challenges and prerequisites of implementing eHealth within OD practice. A topic list was used as a memorandum. This topic list is based on the reports of the aforementioned fifteen group sessions within the pilot about eHealth. 

### 2.5. Data Analysis

All interviews were audio-recorded with the interviewees’ permission. All audio recordings were transcribed verbatim and analyzed using a thematic coding approach [40] with a renowned qualitative data analysis program called Atlas.ti. (www.atlasti.com, accessed on 21 September 2021). Themes were based on the seven OD principles and the purpose of this study (i.e., to determine how eHealth can enhance and be implemented into the OD approach, challenges, and prerequisites). These themes were used as the codes. Fragments of the interviews were coded using the aforementioned codes. When a relevant fragment did not fit within one of these codes, a new code was added. Subsequently, codes were attributed to the three main themes eHealth’s potential value, implementation challenges, and –prerequisites. Any doubts about the coding were discussed with a second researcher. Preliminary results were presented to the OD team to ensure the different perspectives were accurately portrayed and the researcher’s interpretations were trustworthy [41]. The OD team had no further feedback. 

## 3. Results

The results are clustered in three main themes: (1) potential value of eHealth within the transforming OD practice, (2) Challenges related to the use and implementation of eHealth in OD practice, and (3) prerequisites for the implementation of eHealth within OD practice.

### 3.1. The Potential Value of eHealth within the OD Approach

The potential benefits of eHealth within the OD approach were divided into benefits inside and outside the treatment meetings. OD professionals mentioned two applications that could be valuable during treatment meetings: the Ecomap app and video conferencing. The Ecomap app helps to visualize the meaning of relations between network members, which could give useful insights into how relations are experienced within the network or how to increase the network (related to the second OD principle ‘perspective of the social network’). Video conferencing was mentioned as an alternative if a person cannot attend a treatment meeting in person. 

The interviewees believed that eHealth is most beneficial outside the treatment meetings. They mentioned general benefits (such as convenience for clients regarding the time and location of the treatment) and three benefits that are specific to the OD setting—these were improved communication, simplified planning, and broader access to treatment. Improved communication was considered the main benefit of eHealth with regard to (1) the connectedness between the client, their network, and the OD professionals, (2) the possibility of immediate contact with a healthcare professional (related to the first OD principle ‘provide immediate help’), (3) the opportunity to involve network members, (4) ensuring no treatment decisions are made without the client, and (5) the expansion of the social network (all three related to the second OD principle ‘adopt a social network perspective’).

“The most important thing is that there is someone when you need someone” [client].

Even though interviewees see potential benefits of eHealth, all OD professionals reported that they preferred either face-to-face contact alone or face-to-face contact in combination with eHealth. Moreover, all clients said that eHealth cannot substitute face-to-face contact with a healthcare professional.

Besides that, there was some discrepancy between OD professionals and clients regarding continuous and immediate availability. Whereas clients appreciated the idea that OD professionals are available all the time, several OD professionals were reluctant to increase the demands of care. They reported that, like any other medical specialist, they should only be immediately available in crisis situations. They stated that not being immediately available all the time may have a normalizing effect. 

“*There is psychiatry and there is the general hospital. If you have a heart disease or skin problems, then a general practitioner refers you to a specialist. …also for a dermatologist you have a waiting period. And in this situation, we all accept it, then we all consider it as normal. And here it is the opposite. Because here everything should be possible* …”.[OD professional]

To prevent misunderstandings due to these different expectations, interviewees reported that it is helpful to make clear agreements regarding availability, response time, and tool use. These clear agreements also helped to prevent information overflow, which interviewees found stressful.

OD professionals reported that the eHealth planning tool could improve the time-consuming planning of network meetings, by matching agendas from different systems (including an electronic patient record). They also stated that the broad range of protocolled psychoeducation modules in multi-function online treatment programs could clarify any themes that arise during treatment meetings. They also mentioned the added benefit that clients with limited information processing can look back on discussed themes to refresh their memory on what was said. 

### 3.2. Challenges Related to the Use and Implementation of eHealth in OD Practice

Interviewees mentioned several challenges regarding the use and implementation of eHealth in OD practice. First, all clients mentioned that they have no or limited interest and trust in eHealth, and did not ask for eHealth solutions during the treatment meetings. Furthermore, most OD professionals reported limited affinity with eHealth themselves. All interviewees reported a lack of knowledge and experience regarding eHealth and expressed a need to become better informed in using eHealth in OD practice. 

“*I am also not such an eHealth person*”.[professional]

Moreover, most clients regarded multi-function online treatment programs and self-help apps as potentially helpful for others but not for themselves. They said using eHealth was a burden because of the concentration, discipline, self-confidence, and skill needed to express themselves. 

“…*working more personally together with someone. Because if I have to do it on my own, then I get stacked. I do understand it all. If I have to tell it for example on my own to the computer. I don’t know. Somehow, it doesn’t work*”.[client]

Another challenge was the strong conviction of all interviewees that personal contact is a basic need and that care needs face-to-face contact to be effective. 

“…*then you don’t have the one-on-one contact and that is the power of care…If you want to help someone, then you should do that from your heart, then you should do that with love. Otherwise you cannot help the person. That isn’t the case with eHealth. A computer cannot feel love… that’s how it is with eHealth. You never saw me. It is very clinical. A machine*…”.[client]

OD professionals also reported that personal contact with clients at the treatment meetings was vital. They said they needed to be able to respond to the client as a person, using all their senses. Several OD professionals said that sitting together in the same room makes it easier to feel the emotions of the client and their social network members and to see the interactions between network members. They reported that it is more complicated to proceed slowly, adapt to the rhythm of all interviewees, and express empathy nonverbally during online meetings. The clients added that they experienced more support when sitting in the same room than when connected online.

“…*that you really meet someone. The feeling or something. Yes. I think that you feel more, that the other person is really there for you. That it is more special for you or something like that*”.[client]

Another challenge OD professionals expressed is that they feel under pressure to offer eHealth solutions because the organization expects them to use eHealth tools as part of the treatment. Several OD professionals added that they also experience pressure from outside the organization, to ‘solve’ their clients’ and network members’ problems. 

“*And the pressure from outside is high…outside is very broad. Outside is the rest of the organization. Outside is the professional collaboration network. Outside is the news channel which shows every week something about person with confused behavior and things that went wrong. That is the world outside*”.[professional]

According to OD professionals, this pressure makes it harder to adhere to the ‘tolerating uncertainty’ and ‘dialogism’ of OD principles. The eHealth tools were tentatively introduced during reflection moments (principle 2 ‘adopt a social network perspective’, see Table 1), preferably by sharing successful experiences with the eHealth tool rather than giving advice as an expert. Some OD professionals reported the risk of offering an eHealth solution to relieve this pressure; because at least they offered some help to someone who needs it. 

“… *we no longer say all the things we consider. Things emerge in the network. For me, eHealth doesn’t emerge naturally in that context, it emerges because of something else, because I feel forced to introduce it and that pinches*…”.[professional]

Finally, some OD professionals felt that introducing eHealth and the related tools shortly after the OD practice was started presented an additional challenge because starting the OD practice was in itself challenging enough as it requires major changes on multiple levels. Consequently, they did not feel able to actively explore the possibilities of eHealth. 

“…*In the beginning of the process (starting the OD practice), you are mainly occupied with yourself and OD… there is so much to deal with, so much has changed or needs to change*”.[professional]

### 3.3. Prerequisites for Implementation of eHealth within OD Practice

Interviewees mentioned several prerequisites for the implementation of eHealth within OD practice. These prerequisites cover different levels (individual, organizational, product, and societal) and are interactive rather than sequential.

Provided that clients are empowered, interviewees agreed that the most important is that clients must be willing to use eHealth for it to be successfully implemented into their OD practice. Professionals added that they can encourage but not force the use of eHealth.

“*If you are motivated, you are interested and you want it, then it works. But if you think, oh no, I don’t feel like it, then you should actually not start it*”.[client]

“*I don’t get requests from clients anyway…you offer it yourself. And then you need to stimulate it in order to keep it running. And that is not a problem, but at some point they should take up the gauntlet*”.[professional]

Interviewees reported that this willingness of clients to use eHealth depends upon, e.g., knowledge, attitude, skills, and positive experiences with eHealth. They emphasized the relevance of a continuous dialogue about eHealth and to stay connected rather than forcing to use eHealth.

“*Take the time to have open discussions. Explore the resistance. Is there actually resistance?*”.[professional]

Interviewees considered the eHealth group meetings in the pilot a valuable way to learn about and to be enticed to experiment with eHealth tools and mentioned the idea of a learning community.

“*You should introduce people to what is available at the moment. And keep them up-to-date, because it changes rapidly of course. You are together with other people who also don’t know how it works. That helps people to connect…and makes it a bit easier and natural to try it out*”.[professional]

“*Otherwise I would not have tried it. I wouldn’t have done it if someone gave me a flyer or a message like try this out. But now you really talk about it and someone shares his experience. That works better for me*”.[client]

Related to the prerequisites on an organizational level, professionals expressed differing opinions about the extent to which the implementation of eHealth within a client-centered health care service should be voluntary. Some OD professionals explained that board members need to be strong advocates of eHealth and organizational decisions should be made accordingly.

“*If you really find it important. If you as an organization really intrinsically believe that this offers better care, then you need to have the guts to say, ‘we are going to do things radically different*’”.[professional]

Some OD professionals stated that the digital (eHealth) and physical (face-to-face contact) aspects of treatment should be integrated from the start so that clients and OD professionals consider eHealth solutions as part of the treatment. One interviewee explained that this top-down decision would help the OD professionals to justify this substitution to themselves and their clients. Another interviewee believed that OD professionals should be more conscious of the choice of activities and the limited available time. She explained that substituting more face-to-face contacts with online meetings would allow more time for planning treatment meetings, thereby adhering to the principles of ‘immediate help’, ‘social network perspective’, and ‘dialogism’. However, other OD professionals disagreed with a top-down implementation strategy. They believe that OD professionals should offer eHealth solutions reservedly and only mention them if clients express an interest in using them. In their opinion, if clients do not express interest and trust in using eHealth, then forcing eHealth upon them would be disempowering. They believe, like clients do, that the use of eHealth should be voluntary and available as an add-on to face-to-face contacts.

Focusing on the product level, several interviewees said that—if you really want to empower clients—clients should have direct access to and control over the eHealth tools. They also stated that when mental healthcare needs to shift more towards network-oriented care, then the tools should also be more network-oriented, and more tools should be available to relatives.

Moreover, several OD professionals also mentioned that—if we want to consider eHealth as a matter of course within client-centered care—society needs to have more realistic expectations about the role and possibilities of mental health care. They said that power and responsibility need to be transferred to the clients and their social network. Simultaneously, the use of eHealth within OD calls for more self-reliance, and clients should take a more active role in the treatment.

“*Self-reliance could contribute to the use of eHealth*”.[professional]

## 4. Discussion

This study aimed to help the transition towards person-centered and community-based care models by improving our understanding of the value of eHealth within such a transforming mental healthcare setting and to define the challenges and prerequisites for implementing eHealth, in particular within this transforming context. It has been suggested that eHealth could help mental health care models shift from traditional client-clinician roles to more person-centered and community-based healthcare services that empower clients [3,5,6,9,10,13,15]. In face of the current COVID-19 pandemic, this shift may even be regarded as necessary rather than just helpful [42]. Considering their overlapping foundations, one might expect eHealth and a transforming practice such as OD to blend perfectly: both strive for person-centered and network-oriented care, empowerment, connecting people, transparency, and flexibility [10,13,15,17,18,19,30,32,33,43,44,45,46,47,48]. In line with this expectation, professionals suggest that eHealth could support a transforming practice shifting towards more recovery-oriented, person-centered, and community-based service in which shared-decision making is self-evident.

However, the main challenge in the pilot, in which this study took place, is that their clients with severe mental illness were, in general, not motivated to use eHealth. In this study, clients expressed that using eHealth is a burden because of the concentration, discipline, self-confidence, and skill needed to express themselves. This finding is in line with earlier studies that suggest that eHealth tools for persons experiencing serious mental illnesses may require specific design considerations, due to illness-related factors, such as cognitive impairments or mistrust [49,50]. In line with this, most professionals were not strong advocates of the use of eHealth and reported clients’ lack of interest in the use of eHealth as the main issue. They explained that most clients with severe mental illnesses in their caseload don’t cross the threshold to gain experience with eHealth.

This challenge put professionals in the dilemma to what extent eHealth should be used within person-centered care with shared-decision making when the client does not want to use eHealth. To understand this dilemma, it is helpful to delve into underlying assumptions within transforming practices shifting towards more recovery-oriented and person-centered care, such as clients are seen as experts on their own bodies, symptoms, and situations [51], and clients are empowered and participate in care decisions [3,6,7,8,9]. Consequently, the result may be a stalemate, like in chess. Professionals feel the pressure to but cannot use eHealth as long as clients are not willing to use it.

Despite this challenge, professionals agree on the potential value of eHealth within transforming practices. As suggested in previous research [17], our findings support the notion that eHealth could improve communication and thus collaboration between network members. Respecting communication, there was a difference between clients’ desire and professionals’ willingness regarding the online possibility for professionals to be immediately and continuously available. Clients considered this as a benefit, whereas professionals were reluctant to increase the demands of care. Another potential value of eHealth is that it may also facilitate and ensure that clients have an active say in the complete trajectory of their treatment even outside the face-to-face meetings (enabling shared-decision making). In addition, in line with earlier findings [44], our study highlighted the need to simplify the planning of the treatment meetings; organizing meetings with clients and their network members demands more coordination than organizing meetings with individuals. Moreover, in accordance with previous studies [13,18,19], professionals considered the increased accessibility to and scope of healthcare services due to the use of eHealth to be beneficial. Furthermore, videoconferencing was mentioned as a suitable alternative if someone is unable to attend treatment meetings in person, which is in line with earlier findings [52].

Against this background, seeing on one side the potential value of eHealth and on the other side experiencing the lack of interest from clients, the question arose whether a radical top-down decision should be made to make eHealth an established part of a transforming practice. Our interviewees were divided on this topic, clients and some professionals believe that eHealth should evolve bottom-up as a voluntary add-on to face-to-face contacts, and other professionals say that a radical top-down decision to partially substitute face-to-face contacts with eHealth is needed. The COVID-19 pandemic has given us some insight here, as face-to-face contact had to be replaced with online meetings, forcing the digital and physical world to integrate. This has resulted in a drastic increase in the use of eHealth [53]. Whether all contacts should be online unless face-to-face contact is needed remains an open question. Another interesting question that emerges is whether more space could be created for network-oriented treatment meetings if eHealth could replace some face-to-face meetings. The mental health care in ‘the new normal’ will probably consist of more blended care and will need to tolerate diversity as there is no one-fits-all solution [54]. It will also need to be flexible, choosing online treatment when possible and meeting face-to-face when needed [54]. This suggestion fits well with the transforming practice, in which a practice can provide blended care and empowers the client and their network to jointly decide to meet in person when needed.

Zooming in to the threshold of clients and professionals to use eHealth within a transforming practice. The main reason mentioned by all interviewees for the lack of trust and interest in eHealth tools is their strong conviction that personal contact is a basic need and that care needs face-to-face contact to be effective. Their feedback relates to the therapeutic alliance, which is considered vital to the success of face-to-face treatment but remained underexposed in the use of eHealth [55,56]. Merely substituting face-to-face treatment for eHealth interventions may fail to consider the complexity of building up the therapeutic alliance [57]. However, reviews have shown that it is possible for clients to experience therapeutic alliance in digital interventions [56,58,59,60]. Though, more research is needed to gain a better understanding of the specific role of the therapeutic alliance in eHealth tools for individuals experiencing serious mental illnesses, like clients in the current study [57]. In the context of the transforming practice, in which we combine the use of eHealth with the involvement of a network and thus involving more persons, it might be even more complicated to ensure a therapeutic alliance. The experiences in the field of digital systemic practices (e.g., digital family therapy) [61] may give new insights into how to deal with this complexity, even though it is a relatively new study field [62]. For example, research shows that family therapists need to develop complex communication skills to engage with multiple clients through eHealth. Moreover, ethical issues form another level of complexity, such as the question of how digital exclusion influences the involved network members, or for example, the question of whether the choice for using eHealth is equitable across the members of the network.

Another mentioned threshold to use eHealth is a lack of digital skills, which can influence the willingness to start using eHealth [63]. Professionals who feel competent in videoconferencing are more positive about this application [64], suggesting that online contact is more personal to people who are used to it [21]. A recent study suggested that videoconferencing can replace face-to-face meetings more often than previously expected, although customized solutions will be required and whether eHealth is suitable in specific situations will depend on the interplay of multiple factors [54]. Our interviewees emphasized the relevance of a continuous learning process to facilitate dialogue about eHealth and said that attending the group meetings on eHealth within the pilot was a good way to achieve a better understanding of the value of eHealth and to feel competent and confident to use eHealth within the transforming practice. These group meetings helped them to gain experience with eHealth step by step guided by an eHealth expert. This leads to the idea of a reciprocal process in which eHealth facilitates the transforming practice and that the transforming practice facilitates the uptake of eHealth by encouraging clients and their social network to experiment with eHealth, discuss and find a shared understanding and meaning of the changing roles, new design principles, usage possibilities, and limitations of eHealth within a transforming healthcare setting. This process could help to define the ‘the new normal’.

So, there are several challenges with implementing eHealth into a transforming practice, which seems to be solvable based on previous studies. However, in line with several studies [13,14,57,65], our study has shown that an integration of eHealth into a transforming practice involves several interacting prerequisites involving the client, the OD professional, organization, society, and the digital product. Our study has shown that changes are needed on each of these levels for eHealth to be beneficial. The implementation of eHealth will involve, e.g., cultural changes [13,14,65] because it will change workflows and established professional roles [66,67,68], and may disrupt working styles [69]. Acknowledging the complexity of combining this high-impact change within an already transforming context might be the first step towards creating blended client-centered and network-oriented care.

Several limitations of this study must be considered. As a caveat, eHealth was implemented shortly after the OD approach was introduced. As OD was already a substantial change for the team members, they may not have had the capacity to deal with the introduction of eHealth. As a result, most responses were based on limited experiences. One could argue that more experience with eHealth is needed for clients and OD professionals to value it [64]. However, it is not known yet whether the gained experience during the COVID-19 pandemic will inevitably lead to a more positive attitude. Secondly, the external validity may be limited due to the risk of selection bias and the low number of participants. The risk of selection bias lies in the fact that the manager has selected the professionals and the professionals have invited clients. This part of recruitment is not conducted by or controllable for the researcher. Even though the manager and the professionals were familiar with the inclusion criteria and all clients were eligible, it may be that reasons other than the criteria (unconsciously) played a role in the invitation of participants for this study. This risk of selection bias applies to all participants. Moreover, we accepted all clients that were interested in participation, which brings a risk of missing the perspective of clients who did not want to participate. Nevertheless, the clients who participated vary in attitude (positive, neutral, and negative) towards and experience with eHealth. Regarding the low number of participants, despite the low number, data saturation was reached and professionals considered the findings of the study representative for their caseload. Another limitation is that interviewees were all recruited from one mental health care organization, so our results may not be generalizable to the entire target population.

Taken together, the results of this study and lessons learned from the COVID-19 pandemic show that we do not yet fully understand ‘the new normal’ and the role of eHealth in developing client-centered and network-oriented treatment and how to deal with clients’ voices, when professionals see the value of eHealth but clients do not want to start using eHealth. Furthermore, it is still unclear whether healthcare professionals will return to their old treatment approaches after the COVID-19 pandemic. The sustainability of the shift towards blended care depends on multiple individual, social, organizational, and economical factors [70]. The shift towards client-centered and network-oriented care models and towards blended care models are both high-impact changes in themselves. Future research should examine whether and how these shifts could be mutually supportive and lead to sustainable blended client-centered and network-oriented care models.

## Figures and Tables

**Table 1 ijerph-18-10287-t001:** Description of the seven OD principles.

Theme	OD Principles	Description
Treatment organization	**Provide immediate help**	The first treatment meeting is organized within 24 h of the client making contact with an OD professional [33]
	**Adopt a social** **network perspective**	The client’s social network is invited to participate in the treatment meetings from the beginning. Every participant is equally involved in the treatment meeting. Using a relational focus, OD professionals collect all viewpoints and support shared decision-making. The OD adage ‘nothing about me without me’ reflects the transparency of therapy planning and decision-making, which involve all participants [32]. OD professionals share their thoughts with the client and their social network during reflection moments [32].
	**Flexibility and mobility**	The location, frequency, and content of treatment meetings are organized according to the client’s needs [33]. The dialogue proceeds slowly, attuned to the rhythm and needs of each participant.
	**Responsibility**	OD professionals attending the first treatment meeting are the contact person for the client and organize and plan the treatment meetings [33].
	**Psychological continuity**	The same OD professionals remain involved with the network throughout the whole treatment process [33].
Therapeutic process	**Tolerance of uncertainty**	As OD professionals elicit multiple viewpoints, in which network members often have different ideas of the problem, new possibilities arise. However, these possibilities seldom emerge as an unambiguous solution of how to go on [30]. This requires tolerance of uncertainty related to the process and outcome of the treatment, presence in the interaction, and reaction to contingencies rather than relying on pre-planned interventions or goals [32,34].
	**Dialogism**	The therapeutic stance in the treatment meetings fosters dialogue by emphasizing the present moment, responding to clients’ utterances, using open-ended questions and a relational focus. The dialogue is considered as the core healing factor of the OD practice. The therapeutic change is expected to occur through dialogical interactions, rather than through the advice of professionals. Dialogue as a form of psychotherapy is a mutual process that changes the roles of those involved from an interventionist and object to a participant in subject-subject relations [35] as truly human relational beings [30].

## Data Availability

Restrictions apply to the availability of these data. Data was obtained from clients and health care professionals. They have signed an informed consent that stated that the data is only available for the authors of this article. Data will be available from the authors of this article, only with the permission of the participants.

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
