# Peer review of "Integrating eHealth within a Transforming Mental Healthcare Setting: A Qualitative Study into Values, Challenges, and Prerequisites"

_ijerph, 2021, doi:10.3390/ijerph181910287_

Round 1

Reviewer 1 Report

The present study is significant and attractive, addressing an interesting topic that requires lots of attention in this current period. Overall, Is a well-conducted research article with considerable strengths over the health application in the mental health domain. It is well organized, written and structured, and most importantly, there are just a few studies about this topic in literature, making it quite attractive. However, some concerns require to be properly addressed before yet finalised. I include here some comments and suggestions that need to be addressed.

1) Authors wrote in the introduction (Page 1): “This means e.g. that the clinician should be a guide rather than an expert and becomes a guest in the client’s life rather than hosting the client in the clinical domain. At the same time, the client needs to change from a passive listener to an active participant in the treatment process”. That’s a very vague sentence, and at the very beginning of the manuscript. Please, clarify this point!!

2) A more comprehensive definition ed explanation of the eHealth concept must be provided. Is not enough just to mention a connection between the digitalization transformation as the use of eHealth.

3) Can the authors explain a bit more what they meant with “clients”? are they patients? Are they some sort of customers for healthcare services? 

4) the section (2.4. Data collection) is not clear enough. Is not clear to me how and where the topic list was used to identify the themes for the open interviews.

5) Authors have used for the thematic coding the Atlas.ti program. Please provide some reference or link about this software to encourage other researchers to use it, acknowledge its use and prove the usability of this tool. 

6) Is not clear how the results are clustered in three themes. What was the approach to select three themes?

7) In the discussion, would be interesting to underline much more the differences in the perception and expectation of eHealth use between clients and professionals 

Reviewer 2 Report

Thank you for the opportunity of reviewing this article. This is a very interesting article describing a bold research about integrating eHealth within a transforming mental healthcare setting. It describes a qualitative study that, despite some aspects that perhaps could be revised, I think is very interesting to read and offers an interesting perspective.

These are the aspects that I think that could be revised:

— The most important aspect is related to selection bias and methodology. In Participants and recruitment, the authors perhaps should better explain important aspects. This is a qualitative research with some weak points and, due to these features, selection bias is essential and, in this case, also considerable. Therefore, the authors must clearly describe the main features of the population where the sample is obtained. They must also clearly detail the selection criteria and how the patients were chosen. For example, if the patients were chosen randomly. It also seems that professionals sample was a convenience sample. The selection bias is important but also unavoidable. This is a bold research and I think that the conclusions are really interesting, but the external validity is also very limited by the own characteristics of the research. This is not bad at all, it just must be explained. So, a very good description of the sample, and how it was chosen, is essential. It will help to better contextualize the results and the external validity of the conclusions.

— In the same way, in Data Collection, more details could be offered. For example, the description of the Client/network interviews is too short. The authors could explain if all of them were on-site, the number of interviewers, if there was more people helping the person...

— In Materials and Methods/ Context, the authors write that «Open dialogue (OD) – worldwide implemented in several countries - is an example of the aforementioned transforming healthcare service [24]. OD is an innovative person- centered, network-oriented healthcare model within the biological-driven field of psychi- atry. As well as giving a different perception on mental health problems [4], OD radically reorganizes the treatment system as a whole [25,26]. OD organizes care and therapeutic interventions so that primary treatment involves meetings with clients and their social and professional network. The dialogical process, including the multiple viewpoints from the client’s entire network, is the innovative core of OD in psychiatry [27]. The implementation of OD into everyday practice to achieve a more person-centered healthcare model is challenging [28]». This part is interesting but I think that perhaps it could suit better in the Introduction. This is just a suggestion.

— In that same section, the authors write that «The pilot had two main aims. The first aim was to adhere to the seven OD principles (see Table 1A)». This table should appear next, in the text.

— At the beginning of the Discussion section, the authors could show a summary of the main aims of the research (just 1-2 lines).

— The paragraph «This challenge put professionals in a dilemma. To explain this dilemma, we first need to have a look at underlying assumptions within transforming practices shifting towards more recovery-oriented and person-centered care, such as: clients are seen as experts on their own bodies, symptoms and situations [46] and clients are empowered and participate in care decisions [3,6-9]. Consequential, the dilemma here is to what ex- tent eHealth should be used within person-centered care with shared-decision making – in which the empowered clients participate in care decisions – when the client does not want to use eHealth» seems a bit strange. Perhaps the authors could first describe the dilemma, and then show the explanation.

— Making questions seems also a bit strange. For example, «Could eHealth replace some face-to-face meetings, thereby creating space for network-oriented treatment meetings?», perhaps could be rewritten as «An interesting question that emerges is that if eHealth could replace some face-to-face meetings, thereby creating space for network-oriented treatment meetings».

— Some expressions, like «Let’s have a closer look on the threshold of clients», perhaps are too informal.

— In the article, the authors offer some hints of the limitations of the research. Indeed, they highlight some of them in the Limitations section. But I think that they should clearly explain the importance of the selection bias, the low number of participants, and the limitation of the methods used. Of course, that these aspects do considerably limit the external validity of the conclusions. But they do not invalidate the research, and this research is really bold and interesting, and I really liked it. I only think that these limitations must be clearly stated, so the readers can better understand the validity of the conclusions. The authors are bold when they describe the important limitations of eHealth in this particular context, opening new ways for future research.
